# Pseudo-Labeled Auto-Curriculum Learning for Semi-Supervised Keypoint Localization

**Can Wang**[1]    **Sheng Jin**[2,1]    **Yingda Guan**[1]    **Wentao Liu**[1*]    **Chen Qian**[1]
**Ping Luo**[2]    **Wanli Ouyang**[3]

[1]SenseTime Research and Tetras.AI    [2]The University of Hong Kong    [3]The University of Sydney

{wangcan, jinsheng}@tetras.ai   {guanyingda, liuwentao, qianchen}@sensetime.com

pluo@cs.hku.hk   wanli.ouyang@sydney.edu.au

## Abstract

Localizing keypoints of an object is a basic visual problem. However, supervised learning of a keypoint localization network often requires a large amount of data, which is expensive and time-consuming to obtain. To remedy this, there is an ever-growing interest in semi-supervised learning (SSL), which leverages a small set of labeled data along with a large set of unlabeled data. Among these SSL approaches, pseudo-labeling (PL) is one of the most popular. PL approaches apply pseudo-labels to unlabeled data, and then train the model with a combination of the labeled and pseudo-labeled data iteratively. The key to the success of PL is the selection of high-quality pseudo-labeled samples. Previous works mostly select training samples by manually setting a single confidence threshold. We propose to automatically select reliable pseudo-labeled samples with a series of dynamic thresholds, which constitutes a learning curriculum. Extensive experiments on six keypoint localization benchmark datasets demonstrate that the proposed approach significantly outperforms the previous state-of-the-art SSL approaches.

## 1 Introduction

Keypoints (also termed as landmarks) are a popular representation of objects that precisely represent locations of object parts and contain concise information about shapes and poses. Example keypoints are "right shoulder" on a human body or the "tail tip" of a cat. Keypoint localization is the basis of many visual tasks, including action recognition (Yan et al., 2018), fine-grained classification (Gavves et al., 2013; 2015), pose tracking (Jin et al., 2017; 2019) and re-identification (Zhao et al., 2017).

Keypoint localization has achieved great success with the advent of deep learning in recent years (Newell et al., 2016; Xiao et al., 2018; Duan et al., 2019; Sun et al., 2019; Jin et al., 2020a; Xu et al., 2021; Geng et al., 2021; Li et al., 2021b). However, the success of deep networks relies on vast amounts of labeled data, which is often expensive and time-consuming to collect. Semi-supervised learning (SSL) is one of the most important approaches for solving this problem. It leverages extensive amounts of unlabeled data in addition to sparsely labeled data to obtain gains in performance. Pseudo-labeling (PL) has become one of the most popular SSL approaches due to its simplicity. PL-based methods iteratively add unlabeled samples into the training data by pseudo-labeling them with a model trained on a combination of labeled and pseudo-labeled samples.

PL-based methods commonly require a predefined handpicked threshold (Lee et al., 2013; Oliver et al., 2018), to filter out low-confidence noisy predictions. However, a single fixed threshold does not take into account the dynamic capacity of the current model for handling noisy pseudo-labels, leading to sub-optimal performance. In this work, we borrow ideas from Curriculum Learning (CL) (Bengio et al., 2009) and design our curriculum as a series of thresholds for PL, which is tuned according to the feedback from the model. CL is a widely used strategy to control the model training pace by selecting from easier to harder samples. With a carefully designed curriculum, noticeable improvement is obtained. However, traditional CL methods suffer from hand-designed curricula, which heavily rely on expertise and detailed analysis for specific domains. Manual cur-

---

*Corresponding author.

riculum design based on handcrafted criteria is always tedious and sub-optimal. Moreover, curriculum design (or threshold setting) is complicated. High-confidence pseudo-labels typically correspond to easier samples with clean labels, while low-confidence pseudo-labels correspond to harder samples with noisy labels. How to design a curriculum to balance the correctness, representativeness, and difficulty of pseudo-labeled data is an open problem. This paper is devoted to tackling the aforementioned problem, *i.e.* how to automatically learn an optimal learning curriculum for pseudo-labeling in a data-driven way. To this end, we propose a novel method, called Pseudo-Labeled Auto-Curriculum Learning (PLACL). PLACL formulates the curriculum design problem as a decision-making problem and leverages the reinforcement learning (RL) framework to solve it.

Additionally, PL-based methods suffer from confirmation bias (Tarvainen & Valpola, 2017), also known as noise accumulation (Zhang et al., 2016), and concept drift (Cascante-Bonilla et al., 2021). This long-standing issue stems from the use of noisy or incorrect pseudo-labels in subsequent training stages. As a consequence, the noise accumulates and the performance degrades as the learning process evolves over time. To mitigate this problem, we propose the cross-training strategy which alternatively performs pseudo-label prediction and model training on separate sub-datasets.

We benchmark PLACL on six keypoint localization datasets, including LSPET (Johnson & Everingham, 2011), MPII (Andriluka et al., 2014), CUB-200-2011 (Welinder et al., 2010), ATRW (Li et al., 2019c), MS-COCO (Lin et al., 2014), and AnimalPose (Cao et al., 2019). We empirically show that PLACL is general and can be applied to various keypoint localization tasks (human and animal pose estimation) and different keypoint localization networks. With a simple yet effective search paradigm, our method significantly boosts the keypoint estimation performance and achieves superior performance to other SSL methods. We hope our method will inspire the community to rethink the potential of PL-based methods for semi-supervised keypoint localization.

Our main contributions can be summarized as follows:

- We propose Pseudo-Labeled Auto-Curriculum Learning (PLACL). It is an an automatic pseudo-labeled data selection method, which learns a series of dynamic thresholds (or curriculum) via reinforcement learning. To the best of our knowledge, this is the first work that explores automatic curriculum learning for semi-supervised keypoint localization.

- We propose the cross-training strategy for pseudo-labeling to mitigate the long-standing problem of confirmation bias.

- Extensive experiments on a wide range of popular datasets demonstrate the superiority of PLACL over the previous state-of-the-art SSL approaches. In addition, PLACL is model-agnostic and can be easily applied to different keypoint localization networks.

## 2 RELATED WORKS

### 2.1 SEMI-SUPERVISED KEYPOINT LOCALIZATION

Keypoint localization focuses on predicting the keypoints of detected objects, *e.g.* human body parts (Li et al., 2019b; Jin et al., 2020b), facial landmarks (Bulat & Tzimiropoulos, 2017), hand keypoints (Zimmermann & Brox, 2017) and animal poses (Cao et al., 2019). However, training a keypoint localization model often requires a large amount of data, which is expensive and time-consuming to collect. Semi-supervised keypoint localization is one of the most promising ways to solve this problem. Semi-supervised keypoint localization can be categorized into consistency regularization based methods and pseudo-labeling based methods. Consistency regularization methods (Honari et al., 2018; Moskvyak et al., 2020) assume that the output of the model should not be invariant to realistic perturbations. These approaches typically rely on modality-specific augmentation techniques for regularization. Pseudo-labeling methods (Ukita & Uematsu, 2018; Dong & Yang, 2019; Cao et al., 2019; Li & Lee, 2021) use labeled data to predict the labels of the unlabeled data, and then train the model in a supervised way with a combination of labeled and selected pseudo-labeled data. Our approach also builds upon pseudo-labeling methods. In contrast to previous works, we propose to learn pseudo-labeled data selection via reinforcement learning.

## 2.2 CURRICULUM LEARNING

Curriculum learning is firstly introduced by Bengio et al. (2009). It is a training strategy that trains machine learning models from easy to complex samples, imitating human education. The curriculum is often pre-determined by heuristics (Khan et al., 2011; Bengio et al., 2009; Spitkovsky et al., 2009). However, it requires expert domain knowledge and exhaustive trials to find a good curriculum suitable for a specific task and its dataset. Recently, automatic curriculum learning methods are introduced to break through these limits. Popular ones include self-paced learning methods (Kumar et al., 2010; Jiang et al., 2014; Zhao et al., 2015) and reinforcement learning (RL) based methods (Graves et al., 2017; Matiisen et al., 2019; Fan et al., 2018). Our approach can be categorized as RL-based methods. Unlike previous works that focus on supervised learning, our approach is designed for the SSL paradigm. Our work is mostly related to Curriculum Labeling (Cascante-Bonilla et al., 2021). It adopts a *hand-crafted* curriculum based on Extreme Value Theory (EVT) to facilitate model training. In contrast, we propose an automatic curriculum learning approach by searching for dynamic thresholds for pseudo-labeling. In addition, the curriculum of (Cascante-Bonilla et al., 2021) is coarse-grained on the round level, while our curriculum is fine-grained on the epoch level.

## 2.3 REINFORCEMENT LEARNING FOR AUTOML

Reinforcement learning (RL) has shown impressive results in a range of applications. Well-known examples include game playing (Mnih et al., 2015; Silver et al., 2016; 2017) and robotics control(Schulman et al., 2015; Lillicrap et al., 2016). Recent works have employed RL to the AutoML, automating the design of a machine learning system, *e.g.* searching for neural architectures (Zoph & Le, 2017; Zoph et al., 2018; Baker et al., 2017; Pham et al., 2018), augmentation policies (Cubuk et al., 2019), activation functions (Ramachandran et al., 2017), loss functions (Li et al., 2019a; 2021a), and training hyperparameters (Dong et al., 2020). In contrast to these works, we apply RL to the automatic selection of pseudo-labeled data in the context of pseudo-labeling.

## 3 PSEUDO-LABELED AUTO-CURRICULUM LEARNING (PLACL)

### 3.1 OVERVIEW

Our PLACL algorithm is illustrated in Fig. 1. The training process consists of $R$ self-training rounds and each round consists of $N$ training epochs. (0) In the initial round ($r = 0$), we pre-train a keypoint localization network $\Theta_\omega^0$ on the labeled data, where $\omega$ denotes the weights of the network. And for the $r$-th round, (1) The trained network $\Theta_\omega^r$ is used to predict pseudo-labels for unlabeled data. (2) We adopt reinforcement learning (RL) to automatically generate the learning curriculum. Specifically, our curriculum ($\Gamma^r$) consists of a series of thresholds for pseudo-labeled data selection. $\Gamma^r = [\gamma_1^r, \ldots, \gamma_N^r]$, where $\gamma_i^r \in [0, 1]$ is the threshold for each epoch $i$. (3) We then select reliable pseudo-labeled data by the searched curriculum. (4) We retrain a new model ($\Theta_\omega^{r+1}$) using both the labeled samples and selected pseudo-labeled samples. (5) This process is repeated for $R$ rounds.

### 3.2 PSEUDO-LABEL SELECTION FOR SEMI-SUPERVISED KEYPOINT LOCALIZATION

We denote the labeled dataset with $N_l$ samples as $\mathbb{D}_l = \left\{ \left( \boldsymbol{I}_i^l, \boldsymbol{Y}_i^l \right)\big|_{i=1}^{N_l} \right\}$, where $\boldsymbol{I}_i$ and $\boldsymbol{Y}_i$ denote the $i$-th training image and its keypoint annotations (the x-y coordinates of $K$ keypoints). The $N_u$ unlabeled images are denoted as $\mathbb{D}_u = \left\{ \left( \boldsymbol{I}_i^u \right)\big|_{i=1}^{N_u} \right\}$, which are not associated with any ground-truth keypoint labels. Generally, we have $|N_l| \ll |N_u|$.

Pseudo-labeling based method builds upon the general idea of self-training (McLachlan, 1975), where the keypoint localization network $\Theta_\omega$ goes through multiple rounds of training. In the initialization round, the model is first trained with the small labeled training set $\mathbb{D}_{train} = \mathbb{D}_l$ in a usual supervised manner. In subsequent rounds, the trained model is used to estimate labels for the unlabeled data $\tilde{\mathbb{D}}_u = \left\{ \left( \boldsymbol{I}_i^u, \tilde{\boldsymbol{Y}}_i^u \right)\big|_{i=1}^{N_u} \right\}$. Here, we omit the superscript $r$ for simplicity. Specifically, given an unlabeled image $\boldsymbol{I}_i^u$, the trained keypoint localization network $\Theta_\omega$ predicts $K$ heatmaps. Each heatmap is a 2D Gaussian centered on the joint location, which represents the confidence of

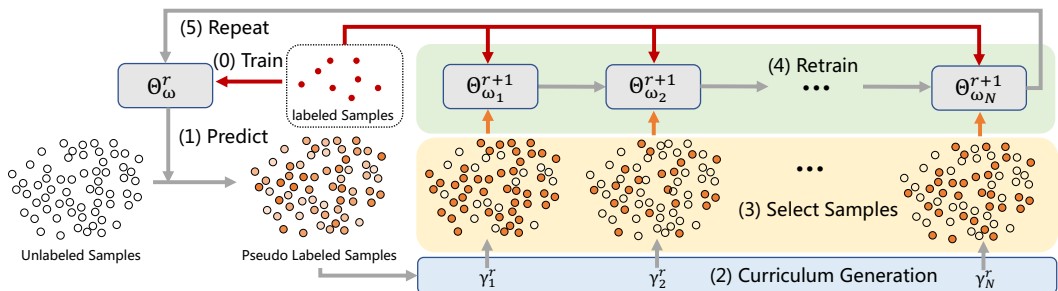

Figure 1: Pseudo-Labeled Auto-Curriculum Learning (PLACL). (0) In the initial round, the model $\Theta_\omega^0$ is pre-trained on the labeled data. And for the $r$-th round, (1) the trained network $\Theta_\omega^r$ is used to predict pseudo-labels for unlabeled data. (2) A learning curriculum consisting of a series of thresholds ($\gamma_i^r$) is generated. (3) Reliable pseudo-labeled data is selected by the searched curriculum. (4) A new model $\Theta_\omega^{r+1}$ is retrained using both the labeled samples and selected pseudo-labeled samples. (5) This process is repeated by re-labeling unlabeled data using the new model.

the $k$-th keypoint. The output pseudo-labeled keypoint location ($\tilde{\boldsymbol{Y}}_i^u$) is the highest response in the heatmap space. And the confidence score $\mathcal{C}\left(\Theta_\omega(\boldsymbol{I}_i^u)\right)$ is the response value at the keypoint location.

Then, pseudo-label selection process is adopted. Let $\boldsymbol{g} = [g_1, ..., g_{N_u}] \subseteq \{0, 1\}^{N_u}$ be a binary vector representing the selection of pseudo-labels, where $g_i$ denotes whether the keypoint prediction on $\boldsymbol{I}_i^u$ is selected.

$$g_i = \left\{ \begin{array}{ll} 1, & \text{if } \mathcal{C}\left(\Theta_\omega(\boldsymbol{I}_i^u)\right) > \gamma \\ 0, & \text{otherwise} \end{array} \right. \tag{1}$$

where $\gamma \in (0, 1)$ is the confidence threshold. Pseudo-labeled samples with higher confidence are added to the training set.

$$\mathbb{D}_{train} = \left\{ \left. \left(\boldsymbol{I}_i^l, \boldsymbol{Y}_i^l\right)\right|_{i=1}^{N_l} \right\} \cup \left\{ \left. \left(\boldsymbol{I}_i^u, \tilde{\boldsymbol{Y}}_i^u\right)\right|_{i=1}^{N_u} \text{ where } g_i = 1 \right\}. \tag{2}$$

Then the keypoint localization network is retrained with a combination of labeled and pseudo-labeled training data $\mathbb{D}_{train}$.

### 3.3 CROSS-TRAINING STRATEGY

In this section, we introduce the cross-training strategy in our curriculum learning framework. In a typical self-training round, the model predicts noisy pseudo-labels which are used in subsequent training stages. Since the pseudo-label prediction is performed on a dataset of *known data* (on which training is performed), the noise accumulates in a positive feedback loop. This causes the long-standing issue of confirmation bias (Tarvainen & Valpola, 2017). To mitigate this problem, we propose the cross-training strategy. Specifically, we randomly partition the unlabeled data $\mathbb{D}_u$ into two complementary subsets $\mathbb{D}_u^{(1)}$ and $\mathbb{D}_u^{(2)}$. The partition is fixed across all the training rounds. The self-training conducts alternatively for $\mathbb{D}_u^{(1)}$ and $\mathbb{D}_u^{(2)}$. For $r = \{1, \ldots, R\}$, when $r$ is odd, we use a combination of $\tilde{\mathbb{D}}_u^{(1)}$ and $\mathbb{D}_l$ to train the model, and the trained model predicts pseudo-labels for the next round on $\mathbb{D}_u^{(2)}$. When $r$ is even, we use a combination of $\tilde{\mathbb{D}}_u^{(2)}$ and $\mathbb{D}_l$ to train the model, and perform pseudo-label prediction on $\mathbb{D}_u^{(1)}$. Before each round, the model parameters are re-initialized with random weights following Cascante-Bonilla et al. (2021) to avoid noise accumulation.

### 3.4 CURRICULUM RESIDUAL LEARNING

Directly learning for $R$ rounds of curriculum parameters separately can be inefficient. Considering that the model is reinitialized and retrained in each self-training round, we assume that the optimal

curricula for different rounds should have some similar patterns. We propose a greedy multi-step searching algorithm. For round $r$, we use the searched curriculum in the previous round $r - 1$ as the base curriculum to guide the searching of current curricula $\Gamma^r$. Inspired by ResNet (He et al., 2016), we propose the curriculum residual learning strategy. Formally, we learn a bias term $\Delta\Gamma^r$ around the base curriculum, *i.e.* $\Gamma^r = (\Gamma^{r-1})^* + \Delta\Gamma^r$, where $(\Gamma^{r-1})^*$ means the searched optimal curriculum for round $r - 1$, and $(\Gamma^0)^*$ is initialized with all zeros. We empirically find that this strategy accelerates the model convergence speed and achieves marginally better performance. To further reduce the search space, we choose every $G$ epochs as an epoch group, which shares the same threshold parameters for pseudo-labeled data selection. In total, we have $N_G$ epoch groups, and the size of each epoch group is $G$. Therefore, the search space is reduced by a factor of $G$, where $G = 10$ in our implementation.

## 3.5 CURRICULUM SEARCH VIA REINFORCEMENT LEARNING

Our PLACL can be formulated as an optimization problem shown in Eq. 3. In the inner-loop, we optimize the weights $\omega$ of the keypoint localization network $\Theta_\omega$ to minimize the training loss L (see Alg. 1). In the outer-loop, we apply proximal policy optimization (PPO2) algorithm (Schulman et al., 2017) to search for the curriculum $\Gamma$ that maximize the evaluation metric $\xi$ (*e.g.* PCK) on the validation set $\mathbb{D}_{\text{val}}$ (see Alg. 2).

$$\max_\Gamma \xi(\Gamma) = \xi(\Theta_{\omega^*(\Gamma)}; \mathbb{D}_{\text{val}}), \quad \text{s.t.} \quad \omega^*(\Gamma) = \arg\min_\omega L(\Theta_\omega; \mathbb{D}_{\text{train}}, \Gamma). \tag{3}$$

The training consists of $R$ rounds. In each round $r$, the PPO2 search process consists of $T$ sampling steps. In each step, $M$ sets of parameters are sampled independently from a truncated normal distribution (Nakano et al., 2012; Fujita & Maeda, 2018), $\Delta\Gamma^r \sim \mathcal{N}_{\text{trunc}[0,1]}\left(\mu_t^r, \sigma^2 I\right)$, where $\mu_t^r$ and $\sigma^2 I$ are the mean and covariance ($\sigma$ is fixed to $0.2$ in practice). These sampled parameters are used to construct $M$ different training curricula for training $M$ keypoint localization networks separately. Then the mean of the distribution is updated by PPO2 algorithm according to the evaluation score of the $M$ networks.

The objective function of PPO2 is formulated in Eq. 4.

$$J\left(\mu^r\right) = \mathbb{E}_\pi\left[\min\left(\frac{\pi_{\mu^r}\left(\Delta\Gamma_j^r; \mu^r, \sigma^2 I\right)}{\pi_{\mu_t^r}\left(\Delta\Gamma_j^r; \mu_t^r, \sigma^2 I\right)}\tilde{\xi}\left(\Gamma_j^r\right), \text{ CLIP }\left(\frac{\pi_{\mu^r}\left(\Delta\Gamma_j^r; \mu^r, \sigma^2 I\right)}{\pi_{\mu_t^r}\left(\Delta\Gamma_j^r; \mu_t^r, \sigma^2 I\right)}, 1 - \epsilon, 1 + \epsilon\right)\tilde{\xi}\left(\Gamma_j^r\right)\right)\right], \tag{4}$$

where the function $\text{CLIP}(x, 1 - \epsilon, 1 + \epsilon)$ clips $x$ to be no more than $1 + \epsilon$ and no less than $1 - \epsilon$. Following the common practice (Li et al., 2021a), the mean reward is subtracted for better convergence. $\tilde{\xi}\left(\Gamma_j^r\right) = \xi\left(\Gamma_j^r\right) - \frac{1}{M}\sum_{j=1}^M \xi\left(\Gamma_j^r\right)$ and the policy $\pi_{\mu^r}$ is defined as the probability density function (PDF) of the truncated normal distribution. PPO2 enforces the *probability ratio* between old and new policies $\pi_{\mu^r}\left(\Delta\Gamma_j^r; \mu^r, \sigma^2 I\right) / \pi_{\mu_t^r}\left(\Delta\Gamma_j^r; \mu_t^r, \sigma^2 I\right)$ to stay within a small interval to control the size of each policy update. We then compute the gradients and update the parameters by $\mu_{t+1}^r \leftarrow \mu_t^r + \alpha\nabla_{\mu^r} J\left(\mu^r\right)$ with a learning rate of $\alpha > 0$. After $T$ sampling steps, we choose $\mu_t^r$ with the highest average evaluation score as $(\Gamma^r)^*$. And $(\Gamma^r)^*$ is used as the base curriculum for the next round. And after $R$ rounds, our final optimal curriculum is obtained, $\Gamma^* = [(\Gamma^1)^*, \ldots, (\Gamma^R)^*]$.

**Training details** In the training phase, the keypoint localization network and the curriculum search policy are simultaneously optimized. For the outer-loop, the PPO2 (Schulman et al., 2017) search procedure is conducted for $T = 16$ sampling steps, and in each step $M = 8$ sets of parameters (curriculum) are sampled. The clipping threshold is $\epsilon = 0.2$, and $\mu_{t+1}^r$ is updated with the learning rate of $\alpha = 0.2$. We empirically use $R = 6$ self-training rounds, and group size $G = 10$ for curriculum search. For the inner-loop, we follow the common practice (Sun et al., 2019; Contributors, 2020) to train the keypoint localization network with Mean-Squared Error (MSE) loss for $N = 210$ epochs per round. Adam (Kingma & Ba, 2015) with a learning rate of $0.001$ is adopted. We reduce the learning rate by a factor of 10 at the 170-th and 200-th epochs. Although the RL search process increases the training complexity, the total training cost is not too high (only 1.5 days with 32 NVIDIA Tesla V100 GPUs). More detailed training settings for each task are provided in §4.2 and §4.3.

Table 1: Keypoint localization with different percentage of labeled images. We report mean and standard deviation from three runs for different randomly sampled labeled subsets. Pseudo-labeling (PL) based methods are not evaluated for 100% of labeled data because there is no unlabeled data to generate pseudo-labels for. The results marked with '*' are from (Moskvyak et al., 2020).

| Method | Percentage of labeled images | | | | |
|---|---|---|---|---|---|
| | 5% | 10% | 20% | 50% | 100% |
| **Dataset 1: LSPET** | | | | | |
| HRNet* (Sun et al., 2019) | 40.19±1.46 | 45.17±1.15 | 55.22±1.41 | 62.61±1.25 | 72.12±0.30 |
| ELT* (Honari et al., 2018) | 41.77±1.56 | 47.22±0.91 | 57.34±0.94 | 66.81±0.62 | 72.22±0.13 |
| Gen* (Jakab et al., 2018) | 61.01±1.41 | 67.75±1.00 | 68.80±0.91 | 69.70±0.77 | 72.25±0.55 |
| SSKL* (Moskvyak et al., 2020) | 66.98±0.94 | 69.56±0.66 | 71.85±0.33 | 72.59±0.56 | 74.29±0.21 |
| PL* (Radosavovic et al., 2018) | 37.36±1.89 | 42.05±1.68 | 48.86±1.23 | 64.45±0.96 | - |
| CL (Cascante-Bonilla et al., 2021) | 61.27±1.54 | 65.43±1.19 | 69.14±0.93 | 70.29±1.18 | - |
| PLACL (Ours) | **70.76±1.47** | **71.91±1.15** | **72.30±0.88** | **72.73±1.23** | - |
| **Dataset 2: MPII** | | | | | |
| HRNet* (Sun et al., 2019) | 66.22±1.60 | 69.18±1.03 | 71.83±0.87 | 75.73±0.35 | 81.11±0.15 |
| ELT* (Honari et al., 2018) | 68.27±0.64 | 71.03±0.46 | 72.37±0.58 | 77.75±0.31 | 81.01±0.15 |
| Gen* (Jakab et al., 2018) | 71.59±1.12 | 72.63±0.62 | 74.95±0.32 | 79.86±0.19 | 80.92±0.32 |
| SSKL* (Moskvyak et al., 2020) | 74.15±0.83 | 76.56±0.48 | 78.46±0.36 | 80.75±0.32 | 82.12±0.14 |
| PL* (Radosavovic et al., 2018) | 62.44±1.75 | 64.78±1.44 | 69.35±1.11 | 77.43±0.48 | - |
| CL (Cascante-Bonilla et al., 2021) | 72.03±1.56 | 73.15±0.95 | 75.80±0.92 | 77.49±0.35 | - |
| PLACL (Ours) | **77.83±1.41** | **78.36±0.92** | **79.68±0.72** | **80.81±0.24** | - |
| **Dataset 3: CUB-200-2011** | | | | | |
| HRNet* (Sun et al., 2019) | 85.77±0.38 | 88.62±0.14 | 90.18±0.22 | 92.60±0.28 | 93.62±0.13 |
| ELT* (Honari et al., 2018) | 86.54±0.34 | 89.48±0.25 | 90.86±0.13 | 92.26±0.06 | 93.77±0.18 |
| Gen* (Jakab et al., 2018) | 88.37±0.40 | 90.38±0.22 | 91.31±0.21 | 92.79±0.14 | 93.62±0.25 |
| SSKL* (Moskvyak et al., 2020) | 91.11±0.33 | 91.47±0.36 | 92.36±0.30 | 92.80±0.24 | 93.81 ±0.13 |
| PL* (Radosavovic et al., 2018) | 86.31±0.45 | 89.51±0.32 | 90.88±0.28 | 92.78±0.27 | - |
| CL (Cascante-Bonilla et al., 2021) | 91.46±0.41 | 92.35±0.34 | 92.74±0.27 | 92.97±0.21 | - |
| PLACL (Ours) | **93.01±0.33** | **93.28±0.29** | **93.45±0.25** | **93.84±0.18** | - |
| **Dataset 4: ATRW** | | | | | |
| HRNet* (Sun et al., 2019) | 69.22±0.87 | 77.55±0.84 | 86.41±0.45 | 92.17±0.18 | 94.44±0.10 |
| ELT* (Honari et al., 2018) | 74.53±1.24 | 80.35±0.96 | 87.98±0.47 | 92.80±0.21 | 94.75±0.14 |
| Gen* (Jakab et al., 2018) | 89.54±0.57 | 90.48±0.49 | 91.16±0.13 | 92.27±0.24 | 94.80±0.13 |
| SSKL* (Moskvyak et al., 2020) | 92.57±0.64 | 94.29±0.66 | 94.49±0.36 | 94.63±0.18 | 95.31±0.12 |
| PL* (Radosavovic et al., 2018) | 67.97±1.07 | 75.26±0.74 | 84.69±0.57 | 92.15±0.24 | - |
| CL (Cascante-Bonilla et al., 2021) | 87.01±1.08 | 89.13±0.94 | 92.34±0.51 | 93.57±0.26 | - |
| PLACL (Ours) | **94.37±0.86** | **94.59±0.80** | **94.85±0.48** | **95.01±0.17** | - |
| **Dataset 5: MS-COCO'2017** | | | | | |
| HRNet (Sun et al., 2019) | 62.44±1.26 | 66.02±1.07 | 69.62±0.84 | 72.81±0.73 | 74.61±0.58 |
| CL (Cascante-Bonilla et al., 2021) | 64.47±1.18 | 67.82±0.95 | 70.36±0.89 | 72.92±0.84 | |
| PLACL (Ours) | **69.39±1.03** | **70.11±0.89** | **71.84±0.66** | **73.42±0.57** | - |

# 4 EXPERIMENTS

## 4.1 DATASETS AND EVALUATION METRICS

**Datasets:** To show the versatility of PLACL, we conduct experiments on 5 diverse datasets. **LSPET** (Leeds Sports Pose Extended Dataset) (Johnson & Everingham, 2010; 2011) consists of images of people doing sports activities. We use 10,000 images from (Johnson & Everingham, 2011) for training and 2,000 images from (Johnson & Everingham, 2010) for validation and testing. **MPII** Human Pose dataset (Andriluka et al., 2014) is a well-known benchmark for human pose estimation. The images are collected from YouTube videos, showing people doing daily human activities. We follow (Moskvyak et al., 2020) to use 10,000 random images from MPII `train` for training, 3,311 images from MPII `train` for validation and MPII `val` for evaluation. **CUB-200-2011** (Caltech-UCSD Birds-200-2011) (Welinder et al., 2010) dataset is a well-known dataset for SSL. It consists of 200 fine-grained bird species with 15 keypoint annotations. We follow (Moskvyak et al., 2020) to split dataset into training (100 categories with 5,864 images), validation (50 categories with 2,958 images) and testing (50 categories with 2,966 images). **ATRW** (Li et al., 2019c) dataset contains

images of 92 Amur tigers captured from multiple wild zoos in challenging and unconstrained conditions. For each tiger, 15 body keypoints are annotated. The dataset consists of 3,610 images for training, 516 for validation, and 1,033 for testing. **MS-COCO'2017** (Lin et al., 2014) is a popular large-scale benchmark for human pose estimation, which contains over 150,000 annotated people. We randomly select 500 images from COCO `train` for validation, the remaining training set (115k images) for training, and COCO `val` (5k images) for evaluation. We use this dataset to validate the applicability of our approach on large-scale data. **AnimalPose** (Cao et al., 2019) dataset contains 5,517 instances of five animal categories: dog, cat, horse, sheep, and cow. It consists of 2,798 images for training, and the 810 images for validation and 1,000 images for testing. We use it to test the generalization and domain-transfer capacity of our proposed method.

**Evaluation Metrics:** **PCK** (Probability of Correct Keypoint): A detected keypoint is considered correct if the distance between the predicted and true keypoint is within a certain threshold ($\alpha l$), where $\alpha$ is a constant and $l$ is the longest side of the bounding box. We adopt PCK@0.1 ($\alpha = 0.1$) for LSPET (Johnson & Everingham, 2011), CUB-200-2011 (Welinder et al., 2010), and ATRW (Li et al., 2019c) datasets. **PCKh** is adapted from PCK, where $l$ is the head size that corresponds to 60% of the diagonal length of the ground-truth head box. We adopt PCKh@0.5 ($\alpha = 0.5$) for MPII (Andriluka et al., 2014) dataset. Standard **AP** (Average Precision) is another commonly used evaluation metric. It is based on object keypoint similarity (OKS), which measures the distance between predicted keypoints and ground-truth keypoints normalized by the scale of the object. We use mAP for AnimalPose (Cao et al., 2019) datasets.

## 4.2 COMPARISONS WITH THE STATE-OF-THE-ART SSL APPROACHES

In Table 1, we compare with the supervised baseline (HRNet (Sun et al., 2019)) and other state-of-the-art SSL approaches. We experiment with different percentages of labeled images (5%, 10%, 20%, 50%, and 100%). For fair comparisons, all results are obtained using HRNet-w32 backbone with the input size of $256 \times 256$. We follow (Moskvyak et al., 2020) to prepare datasets and exclude half body transforms, and testing tricks (post-processing, and flip testing).

**Comparisons with consistency regularization methods.** ELT (Honari et al., 2018) (equivariant landmark transformation) loss encourages the model to output keypoints that are equivariant to input transformations. Gen (Jakab et al., 2018) learns to extract geometry-related features through conditional image generation. SSKL (Moskvyak et al., 2020) learns the pose invariant keypoint representations with semantic keypoint consistency constraints. These consistency regularization methods have shown superior results over the supervised baseline, however, they are inferior to our PLACL method on all datasets and different percentages of labeled samples. Especially we show that PLACL is mostly effective for low data regimes. For example, in CUB-200-2011 dataset, PLACL with only 5% labeled data achieves better performance (93.01 vs 92.80) than SSKL with 50% labeled data. And in LSPET dataset, we show that PLACL improves the performance of baseline by a large margin from 40.19 to 70.76 with 5% labeled images.

**Comparisons with pseudo-labeling method.** We also compare with a pseudo-labeling (PL) baseline (Radosavovic et al., 2018). Overall PLACL significantly outperforms the PL baseline on all datasets. As pointed out by Moskvyak et al. (2020), the vanilla PL approach does not perform well for the keypoint localization task with a low data regime, due to the lack of an effective pseudo-label selection scheme. Instead, PLACL is able to automatically select high-quality pseudo-labeled samples, which is the key to the success of pseudo-labeling based methods.

**Comparisons with curriculum-learning method.** Curriculum Labeling (CL) (Cascante-Bonilla et al., 2021) is a recently proposed approach that applies a *hand-crafted* curriculum to facilitate training of SSL. We observe that our proposed PLACL significantly outperforms CL, which validates the effectiveness of our proposed *automatic* curriculum learning.

**Experiments on large-scale datasets.** Inspired by Zhou et al. (2020), in order to decrease the RL curriculum search cost for the large-scale MS-COCO (Lin et al., 2014) and full MPII (Andriluka et al., 2014) datasets, we use a light proxy task with reduced number of training samples (5k) for RL curriculum search. After the search procedure, we re-train the keypoint networks with the searched curriculum on the full training set and evaluate them on the test set. Please refer to A.4 for more analysis about proxy tasks and A.5 for experiments on the full MPII (Andriluka et al., 2014) dataset.

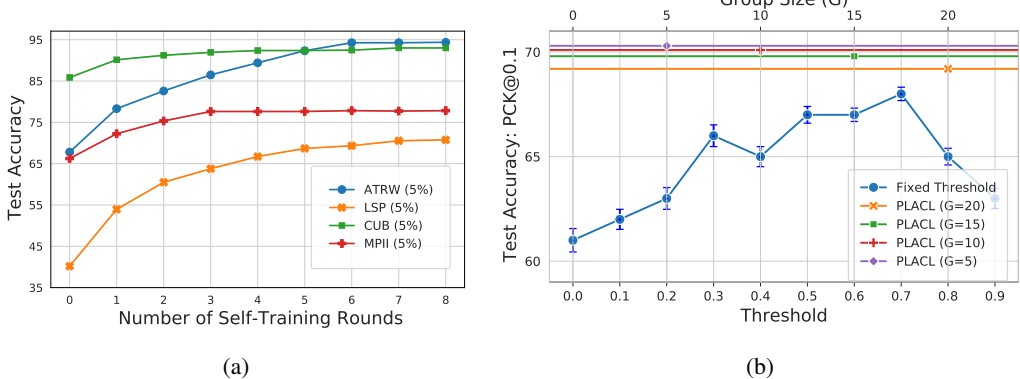

(a)                                                              (b)

Figure 2: (a) Accuracy on different datasets (5% labeled data) with various rounds ($R$). (b) Comparisons with different fixed thresholds (blue dots) and different group sizes $G$. Experiments are conducted on LSPET dataset (5% labeled data).

## 4.3 EVALUATION OF GENERALIZATION CAPACITY

**Generalization to different domains.** We investigate the generalization ability of our proposed model on domain transfer. To this end, we conduct experiments on AnimalPose (Cao et al., 2019) dataset in Table 2. Specifically, we follow (Cao et al., 2019) to choose one animal class (e.g. cat) as the target domain and the remaining four classes for the source domain. The images of the source domain are fully annotated, while the images of the target domain are unlabeled. For fair comparisons, we adopt the AlphaPose model (Fang et al., 2017), which uses ResNet-101 (Xiao et al., 2018) as the backbone. All models are trained with the pose-labeled human dataset involved. The AlphaPose baseline model is pre-trained on the human dataset and fine-tuned on the labeled source animal data. For pseudo-labeling based approaches (Inoue et al., 2018; Cao et al., 2019), the unlabeled target animal data is used for pseudo-labeling. For domain adaptation approaches (Tzeng et al., 2015; Long et al., 2016), the unlabeled target data is used for domain transfer. Please refer to Cao et al. (2019) for details about the compared approaches. We observe that our proposed approach consistently outperforms the previous state-of-the-art methods on cross-domain semi-supervised learning.

Table 2: **Dataset 6: AnimalPose.** Evaluation of generalization capacity to the target unseen animal class. All results are obtained using the AlphaPose model (Fang et al., 2017) with ResNet-101 (He et al., 2016) as the backbone. Results marked with '*' are from Cao et al. (2019).

| Method | mAP for each class | | | | |
|---|---|---|---|---|---|
| | cat | dog | sheep | cow | horse |
| AlphaPose Baseline* (Fang et al., 2017) | 37.6 | 37.3 | 49.4 | 50.3 | 47.9 |
| Dom Confusion* (Tzeng et al., 2015) | 38.0 | 37.7 | 49.5 | 50.6 | 48.5 |
| Residual Transfer* (Long et al., 2016) | 37.8 | 38.2 | 49.1 | 50.8 | 48.6 |
| CycleGAN+PL* (Inoue et al., 2018) | 35.9 | 36.7 | 48.0 | 50.1 | 48.1 |
| WS-CDA+PPLO* (Cao et al., 2019) | 42.3 | 41.0 | 54.7 | 57.3 | 53.1 |
| PLACL (Ours) | **47.1** | **42.9** | **59.5** | **58.4** | **66.0** |

## 4.4 ANALYSIS

**Number of self-training rounds.** Along with the increasing of self-training rounds ($R$), the quality of the pseudo-labels gradually improves (see Fig. 3) and the test accuracy increases (see Fig. 2a) until saturation. The experiments are conducted on multiple datasets with 5% labeled data. Interestingly, different datasets require a different number of rounds to achieve optimal, because four-legged animals (ATRW (Li et al., 2019c)) have more pose variations than birds (CUB-200-2011 (Welinder et al., 2010)). We use $R = 6$, because further increasing $R$ does not bring significant gains.

**Comparisons with pseudo-labeling with different fixed thresholds.** As shown in Fig. 2b, we compare our PLACL with 10 static thresholds from 0.0 to 0.9. We observe that PLACL clearly outperforms all these fixed threshold alternatives. Moreover, we find that the accuracy will be significantly affected by different thresholds (61.01% PCK for $\gamma = 0.0$ vs 67.35% PCK for $\gamma = 0.7$).

**Choice of epoch group size**. In Fig. 2b, we also compare the performance of different epoch group sizes ($G$). We empirically find that smaller $G$ will produce better performance, but at the cost of increased search space. We choose $G = 10$ to trade-off between accuracy and efficiency.

## 4.5 ABLATION STUDIES

In Table 3, we present ablation studies to measure the contribution of each component. All ablative experiments are conducted on LSPET (Johnson & Everingham, 2011) dataset with 5% labeled data. PCK@0.1 is adopted as the evaluation metric.

**Effect of cross-training strategy.** We find that without cross-training strategy the accuracy significantly drops from 70.76 to 67.13, due to noise accumulation over time. Especially, we find that there is little to no improvement after the first self-training round.

**Effect of curriculum learning.** We compare PLACL with the alternative that only searches for a fixed threshold via RL. We find that using dynamic thresholds improves upon the fixed-threshold alternative by a large margin, which validates the effectiveness of curriculum learning.

**Effect of parameter search.** We compare PPO2 (Schulman et al., 2017) search with *Random Search* (w/o PPO2 search). We randomly sampled $T \times M$ curricula and pick out the best one for comparisons. PPO2 search obtains much better performance (70.76 vs 65.42). This indicates that the searching problem is non-trivial and that our searching algorithm is very effective.

We also compare with manually designed curricula whose thresholds are gradually decreasing on the epoch level. We tried five curricula with different decrease slopes and reported the best one as *Manually Design*. We observe that PLACL significantly outperforms the manually-designed curriculum baseline which validates the effectiveness of automatic curriculum search.

Table 3: Ablation studies on LSPET dataset with 5% labeled data.

| Method | PCK@0.1 |
|---|---|
| PLACL, w/o cross-training strategy | 67.13 |
| PLACL, w/o curriculum learning | 68.51 |
| PLACL, w/o PPO2 search (Random Search) | 65.42 |
| PLACL, w/o PPO2 search (Manually Design) | 65.71 |
| PLACL, full method | 70.76 |

## 5 CONCLUSIONS

We propose a novel Pseudo-labeled Auto-Curriculum Learning (PLACL) for the task of semi-supervised keypoint localization. We propose to learn a curriculum to automatically select reliable pseudo-labels and propose cross-training strategy to mitigate the confirmation bias problem. Extensive experiments on 6 diverse datasets validate the effectiveness and versatility of the proposed method. We believe that our proposed approach is generic and we plan to investigate the applicability of PLACL on other visual tasks, such as object detection and semantic segmentation.

**Acknowledgement.** We thank Lumin Xu and Wang Zeng for their valuable feedback to the paper. Ping Luo is supported by the General Research Fund of HK No.27208720 and 17212120. Wanli Ouyang was supported by the Australian Research Council Grant DP200103223, FT210100228, and Australian Medical Research Future Fund MRFAI000085.

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

## A APPENDIX

### A.1 PSEUDO-CODE FOR PLACL ALGORITHM

Here we present the pseudo-code for the proposed Pseudo-Labeled Auto-Curriculum Learning (PLACL) algorithm. In PLACL, the keypoint localization network and the curriculum policy are jointly optimized. In the inner-loop, we optimize the keypoint localization network as shown in Algorithm 1. In the outer-loop, the curriculum policy is updated according to the performance of the keypoint localization network, as shown in Algorithm 2.

---

**Algorithm 1:** Inner-loop network training

---

**Input:**
  1. Labeled data $\mathbb{D}_l$, pseudo-labeled data $\tilde{\mathbb{D}}_u$.
  2. Number of training groups per round $N_G$;
  3. Curriculum $\Gamma$;
**Output:** Obtained optimal network weights $\omega^*(\Gamma)$.

Random initialization of network weights ;
**for** $g = 1$ **to** $N_G$ **do**
  Update current threshold $\gamma_g = \Gamma[g]$ ;
  Compute data selection vector $\boldsymbol{g}$ (Eq. 1) ;
  Construct the training set $\mathbb{D}_{train}$ (Eq. 2) ;
  Update network weights $\omega$ via back-propagation;
**return** $\omega^*(\Gamma)$ ;

---

---

**Algorithm 2:** Pseudo-Labeled Auto-Curriculum Learning (PLACL)

---

**Input:**
  1. Labeled data $\mathbb{D}_l$, unlabeled data $\mathbb{D}_u$.
  2. Initialized distribution $\mu$ and $\sigma^2$.
  3. Number of rounds $R$.
  3. Searching steps $T$; Sampling number $M$.
  4. Evaluation metric (e.g. PCK@0.1) $\xi$.
**Output:** Obtained the optimal curriculum $\Gamma^*$ and the final network $\Theta_{\omega^*}$.

Initialize $(\Gamma^0)^*$ with all zeros.
Pre-train keypoint localization network $\Theta_\omega^0$ using labeled data $\mathbb{D}_l$.
**for** $r = 1$ **to** $R$ **do**
  **if** $r\%2 == 1$ **then**
    Predict keypoint pseudo-labels $\tilde{\mathbb{D}}_u^{(1)}$ with $\Theta_{\omega^*}^{r-1}$;
    $\tilde{\mathbb{D}}_u = \tilde{\mathbb{D}}_u^{(1)}$
  **else**
    Predict keypoint pseudo-labels $\tilde{\mathbb{D}}_u^{(2)}$ with $\Theta_{\omega^*}^{r-1}$;
    $\tilde{\mathbb{D}}_u = \tilde{\mathbb{D}}_u^{(2)}$
  **for** $t = 1$ **to** $T$ **do**
    **for** $j = 1$ **to** $M$ **do**
      Sample parameter $\Delta\Gamma_{j,t}^r \sim \mathcal{N}_{\text{trunc}[0,1]}\left(\mu_t^r, \sigma^2 I\right)$ ;
      $\Gamma_{j,t}^r = \Delta\Gamma_{j,t}^r + (\Gamma^{r-1})^*$ ;
      Get $\Theta_{\omega^*,j,t}^r$ via inner-loop network training using $\Gamma_{j,t}^r$ (Alg. 1) ;
      Compute the evaluation metric $\xi\left(\Gamma_{j,t}^r\right) = \xi(\Theta_{\omega^*,j,t}^r; \mathbb{D}_{\text{val}})$ ;
    Compute the objective function $J(\mu)$ (Eq. 4);
    Update $\mu_{t+1}^r \leftarrow \mu_t^r + \alpha\nabla_{\mu^r}J(\mu^r)$ ;
  $(\Gamma^r)^* = \arg\max_{\mu_t^r} \sum_{j=1}^M \xi\left(\Gamma_{j,t}^r\right), \forall t = 1, \ldots, T$ ;
  Get $\Theta_{\omega^*}^r$ via network training using $(\Gamma^r)^*$ (Alg. 1) ;
**return** $\Gamma^* = [(\Gamma^1)^*, \ldots, (\Gamma^R)^*]$ and $\Theta_{\omega^*}^R$ ;

---

## A.2 VISUALIZATION OF PSEUDO-LABELED SAMPLES

In order to provide a better illustration of how pseudo-labels evolve in self-training rounds, we visualize some pseudo-labeled samples for different datasets. We observe that the quality of pseudo-labels gradually improves with the increase of the self-training rounds.

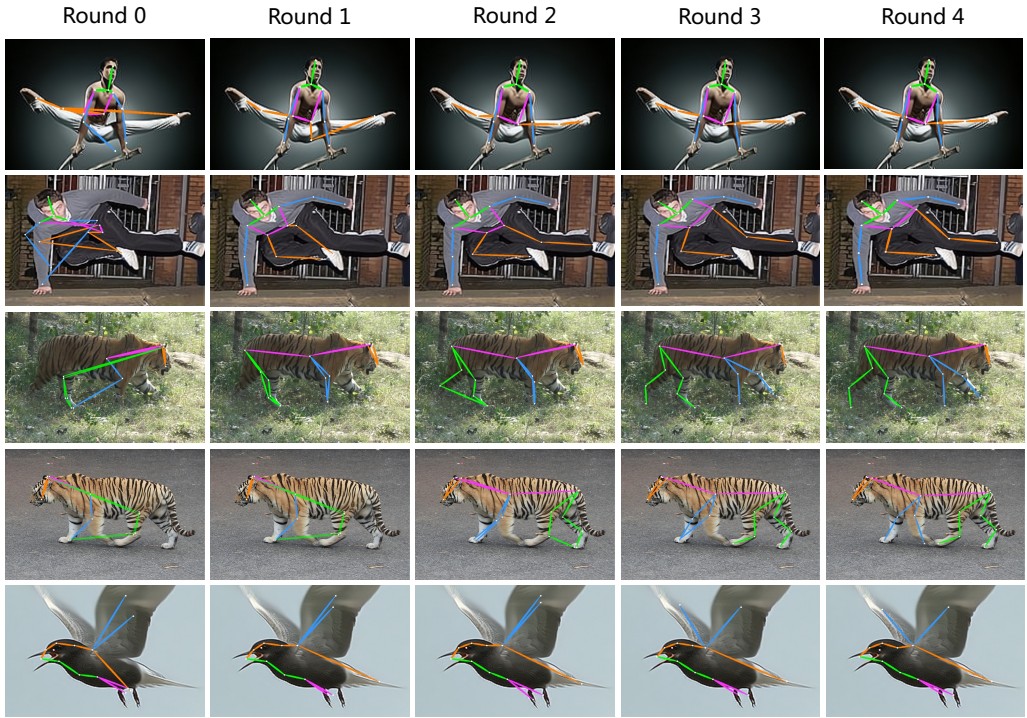

Figure 3: Visualization of how pseudo-labels evolve in self-training rounds.

## A.3 GENERALIZATION TO DIFFERENT KEYPOINT LOCALIZATION MODELS.

Table 4 shows the improvement when PLACL is applied to the recent state-of-the-art keypoint localization models which vary in model architectures and training/testing techniques. The experiments are conducted on LSPET dataset with 5% labeled images and 95% unlabeled images. We show that PLACL consistently improves the performance of the state-of-the-art approaches by a large margin. PLACL does not require any knowledge of the keypoint localization models, making it easy to use in practice.

Table 4: Performance improvement of different keypoint localization methods by PLACL. Experiments are conducted on CUB-200-2011 dataset (5% labeled data) with PCK@0.1 as the metric.

| Method | Backbone | w/o PLACL | w/ PLACL |
|---|---|---|---|
| SimpleBaseline (Xiao et al., 2018) | ResNet-50 | 79.16 | 93.51 |
| SimpleBaseline (Xiao et al., 2018) | ResNet-101 | 81.34 | 93.66 |
| SimpleBaseline (Xiao et al., 2018) | ResNet-152 | 86.15 | 94.27 |
| HRNet (Sun et al., 2019) | HRNet-w32 | 85.86 | 93.01 |
| HRNet (Sun et al., 2019) | HRNet-w48 | 85.89 | 94.26 |
| DARK (Zhang et al., 2020) | HRNet-w32 | 86.67 | 94.18 |

## A.4 ANALYSIS OF PROXY TASKS

There are a lot of methods that target at improving the searching efficiency in literature, *e.g.* using reduced-training proxy tasks (input size, model size, training samples, and training epochs can be reduced (Zhou et al., 2020)). With these techniques, we are able to get orders of magnitude less computation cost, but can still match the performance. Therefore, we believe that the scalability is not a problem. For example, in order to further reduce the complexity, we use a light proxy task (with reduced training samples) for the RL search process. Specifically, we randomly select a small proportion of the training data (*e.g.* 5k images) for efficient curriculum search. After the search procedure, we re-train the keypoint networks with the searched curriculum on the full training set and evaluate them on the test set.

As shown in Table 5, we randomly select different number of training images (5k and 10k) for RL curriculum search. We find that reducing the number of training images by half (from 10k to 5k) does not decrease the final performance much, which validates the effect of using proxy tasks. Such a strategy enables us to apply the proposed PLACL approach to large-scale datasets, such as MS-COCO (Lin et al., 2014) and the full MPII (Andriluka et al., 2014) datasets.

Table 5: We randomly select different number of training images (#Images) for RL curriculum search, and re-train the keypoint networks with the searched curriculum on the full training set.

| #Images | 5% | 10% | 20% | 50% | 100% |
|---|---|---|---|---|---|
| **LSPET** | | | | | |
| 5k | 70.72±1.49 | **71.93±1.17** | 72.24±0.82 | 72.71±1.19 | - |
| 10k | **70.76±1.47** | 71.91±1.15 | **72.30±0.88** | **72.73±1.23** | - |
| **MS-COCO'2017** | | | | | |
| 5k | **69.39±1.03** | 70.11±0.89 | **71.84±0.66** | 73.42±0.57 | - |
| 10k | 69.24±1.02 | **70.12±0.87** | 71.61±0.63 | **73.43±0.61** | - |

## A.5 EXPERIMENTS ON THE FULL MPII DATASET

As shown in Table 6, we provide the results on the full MPII (Andriluka et al., 2014) dataset. Since the codes of SSKL (Moskvyak et al., 2020) are not publicly available, we only compare with CL (Cascante-Bonilla et al., 2021) in the experiments. We see that our proposed PLACL consistently outperforms CL, especially for low labeled data regime (5% and 10%). Note that our results on the full MPII are obtained using reduced-training proxy tasks, *i.e.* we use 5K images for RL curriculum search, and re-train the model with the obtained curriculum on the full training set.

Table 6: Comparisons with CL on the full MPII dataset.

| Method | 5% | 10% | 20% | 50% | 100% |
|---|---|---|---|---|---|
| **Full MPII** | | | | | |
| HRNet (Sun et al., 2019) | 78.00±1.35 | 81.89±0.94 | 82.94±0.67 | 88.34±0.45 | 89.76±0.17 |
| CL (Cascante-Bonilla et al., 2021) | 80.38±1.31 | 83.06±0.89 | 84.57±0.68 | 88.72±0.34 | |
| PLACL (Ours) | **82.21±1.22** | **85.42±0.85** | **86.24±0.56** | **89.16±0.21** | - |

