# OpenReview forum: "Pseudo-Labeled Auto-Curriculum Learning for Semi-Supervised Keypoint Localization"
_ICLR.cc/2022/Conference — ICLR 2022 Poster_

### Official Review · Reviewer_UzKw · 2021-10-29

**Correctness:** 4
**Technical Novelty And Significance:** 2
**Empirical Novelty And Significance:** 2
**Recommendation:** 5
**Confidence:** 4

**Main Review:**

## Strengths

1. The idea of performing ACL using PPO2 is simple and elegant.
2. The paper is well written and easy to read.
3. The method is always either on par or outperforms the keypoint localization network competitors.

## Weaknesses/Questions

1. PLACL seems computationally very intensive. In the experiments M=8, R=6 and T=16, so it means that 736 networks are trained from scratch, isn't it ? If I am not mistaken I did not see any information concerning the training time of PLACL, type and number of GPUs used, and a comparison against sota methods. For instance, how does PLACL compare against SSKL in terms of training time, memory footprint, etc.?
2. The results that are reported in the experiments (Table 1 and 2) for the line "PLACL (ours)" correspond to a forward pass in your implementation of HRNet using which weights? In Alg. 2, the output is the "optimal curriculum". Does it mean that a final training is performed using the last threshold of the optimal curriculum? If that is the case, I suggest to add this step at the end of Alg.2 and return the weights of this network.
3. PLACL seems application agnostic. The results are always either on par or outperform the keypoint localization network competitors but the improvement w.r.t. SSKL for instance is not huge. I suggest to consider a second application to strengthen the paper.
4. In section 4.2, it is said "However, the analysis of their combined effects is outside the scope of this work.". If there exists methods (with codes available) that combine these effects, I suggest to include them in the experiments, even if they outperform PLACL.
5. In Alg.2, pseudo-labels are predicted using $N_{\omega^*}^{r-1}$, is it the network obtained at the previous round for j=M and t=T or do you train another network with ($\Gamma^{r-1})^*$ ? I think this question is related to my remark 2.

**Summary Of The Paper:**

A semi-supervised learning method (PLACL) is proposed. This method employs a Pseudo-Labeling (PL) approach. It consists in iteratively (these iterations are called rounds):

1. predicting pseudo-labels to unlabeled data using the current model,
2. training a series of models from scratch using the labeled data and selections of pseudo-labeled data. The pseudo-label selection is performed using a series of thresholds (called curriculum) over the scores output by the model.

The authors propose an Auto-Curriculum Learning (ACL) strategy to automatically update the curriculum using a reinforcement learning approach (PPO2). The fact that the curriculum is updated at each round is called "Curriculum Residual Learning". They also employ a cross-training strategy to prevent the issue of confirmation bias.

The performances of PLACL are evaluated on a keypoint localization application on 5 different datasets. When the percentage of labeled data is very low (5% or 10%), PLACL outperforms the state of the art semi-supervised learning competitors.


**Summary Of The Review:**

The paper is interesting and well written but the novelty is limited (using PPO2 to update the thresholds), the results are not very impressive (especially compared to SSKL) and the evaluation is limited to a single application (keypoint localization) while PLACL is application agnostic. Thus I believe the paper is not ready for a publication at ICLR. Please answer my questions from the "main review".

---

> ### Author Response · Authors · 2021-11-20
> **To Reviewer UzKw (Part1)**
>
>
> Thanks for the valuable comments. We have addressed your concerns and conducted additional experiments. The paper is revised accordingly with all the requested results added. We clarify the questions as follows.
>
> **The novelty is limited (using PPO2 to update the thresholds)**
>
> Kindly note that ''using PPO2 to update the thresholds'' is *not* our claimed contributions. The major contributions of this paper are (1) to explore automatic curriculum learning for semi-supervised keypoint localization, and (2) apply RL to tackle pseudo-label sample selection. And PPO2 is only one of the techniques to achieve this goal.
>
> **Concerns about time complexity**
> Q1: The information concerning the training time of PLACL, type and number of GPUs used, and a comparison against sota methods.
>
> A1: Although the RL search process indeed increases the training complexity, the total training cost is not too high (only 1.5 days with 32 NVIDIA Tesla V100 GPUs).
>
> There are a lot of methods that target at improving the searching efficiency in literature, *e.g.* sample-efficient reinforcement learning [1], and reduced-training proxy tasks (input size, model size, training samples, and training epochs can be reduced) [2].
> With these techniques, we are able to get orders of magnitude less computation cost, but can still match the performance. Therefore, we believe that the scalability is NOT a problem.
> For example, in order to further reduce the complexity, we use a light proxy task (with reduced training samples) for the RL search process. Specifically, we randomly select a small proportion of the training data (*e.g.* 5k images) for efficient curriculum search. After the search procedure, we re-train the keypoint networks with the searched curriculum on the full training set and evaluate them on the test set.
>
> As shown in Table R1 , we randomly select different number of training images (5k and 10k) for RL curriculum search. We find that reducing the number of training images by half (from 10k to 5k) does not decrease the final performance much, which validates the effect of using proxy tasks.
> Such a strategy enables us to apply the proposed PLACL approach to large-scale datasets, such as MS-COCO'2017 and full MPII dataset.
>
> **Table R1: We perform RL curriculum search on different numbers of training images (\#Images), and re-train the keypoint networks with the searched curriculum on the full training set.**
>
> | Dataset    | #Images | 5% | 10% | 20% | 50% |
> | :--------  |  :--:   | :--: |  :--: | :--: | :--: |
> | LSPET      | 5k      | 70.72$\pm$1.49 | **71.93$\pm$1.17** | 72.24$\pm$0.82 | 72.71$\pm$1.19 |
> | LSPET      | 10k     | **70.76$\pm$1.47**  | 71.91$\pm$1.15  | **72.30$\pm$0.88**  | **72.73$\pm$1.23**  |
> |MS-COCO'2017| 5k      | **69.39$\pm$1.03**  | 70.11$\pm$0.89  | **71.84$\pm$0.66**  | 73.42$\pm$0.57  |
> |MS-COCO'2017| 10k     | 69.24$\pm$1.02 | **70.12$\pm$0.87**  | 71.61$\pm$0.63 | **73.43$\pm$0.61** |
> | | |
>
> [1] Buckman, Jacob, et al. "Sample-efficient reinforcement learning with stochastic ensemble value expansion." arXiv preprint arXiv:1807.01675 (2018).
>
> [2] Zhou, Dongzhan, et al. "Econas: Finding proxies for economical neural architecture search." Proceedings of the IEEE/CVF Conference on Computer Vision and Pattern Recognition. 2020.
>
>
> **Q2: The results that are reported in the experiments (Table 1 and 2) for the line "PLACL (ours)" correspond to a forward pass in your implementation of HRNet using which weights? In Alg. 2, the output is the "optimal curriculum". Does it mean that a final training is performed using the last threshold of the optimal curriculum? If that is the case, I suggest to add this step at the end of Alg.2 and return the weights of this network.**
>
> A2: Thanks for pointing out this! Actually, the model training is performed after obtaining the optimal curriculum at each round. Finally, Alg.2 returns both the optimal curriculum and the weights of this network at the same time. We have revised the paper (especially Alg.2) accordingly.
>
> (1) The line "PLACL (ours)" corresponds to the model $\Theta_{\omega^*}^{R}$ produced by the final round $R$. (Note that the keypoint network is now represented by $\Theta$ in the revised version.)
>
> (2) Thanks for the suggestion! Actually, the model is re-trained at each round. We have added this step in Alg.2.

---

> > ### Author Response · Authors · 2021-11-20
> > **To Reviewer UzKw (Part2)**
> >
> >
> >
> > **Q3: The results are always either on par or outperform the keypoint localization network competitors but the improvement w.r.t. SSKL for instance is not huge.**
> >
> > A3: We would like to respectively point out that our propose PLACL demonstrates superior performance to the state-of-the-art semi-supervised baselines, especially in low labeled data regime (5% and 10% of labeled data). As the amount of labeled data increases, our method gets close to the upper limit (100% supervised learning baseline). Kindly note that SSKL belongs to the consistency regularization methods, while our PLACL belongs to the pseudo-labeling based methods. They are complementary to each other. We also show that our proposed PLACL significantly outperforms the state-of-the-art pseudo-labeled baselines, which already validates the effectiveness of our proposed method.
> >
> >
> > **Q4: PLACL seems application agnostic. I suggest to consider a second application to strengthen the paper.**
> >
> > A4: Thanks for the great suggestion! It is really exciting to investigate the applicability of PLACL on other visual tasks. We plan to conduct experiments on other applications such as object detection and semantic segmentation in the future.
> >
> > Keypoint localization is an important research topic, covering a wide range of sub-topics, such as human pose estimation and animal behavior analysis. In this work, we have already conducted extensive experiments on six different datasets (including the newly added MS-COCO dataset). We believe it is sufficient to validate the generalizability of our proposed method.
> >
> >
> >
> >
> > **Q5: In section 4.2, it is said "However, the analysis of their combined effects is outside the scope of this work.". If there exists methods (with codes available) that combine these effects, I suggest to include them in the experiments, even if they outperform PLACL.**
> >
> > A5: Sorry for the misunderstanding. In fact, to the best of our knowledge, there do not exist such approaches that combine these effects for semi-supervised keypoint localization. But we plan to explore this promising direction in the future. We have revised the paper to make it clear.
> >
> > **Q6: In Alg.2, pseudo-labels are predicted using $\rm{N}_{\omega^{\*}}^{r-1}$. Do you train another network with $(\Gamma^{r-1})^\*$?**
> >
> > A5: Yes, we re-train the network $\rm{N}_{\omega^{\*}}^{r-1}$ with the obtained $(\Gamma^{r-1})^\*$. We have revised the paper to make it clear.

---

> > > ### Comment · Reviewer_UzKw · 2021-11-29
> > > **Response**
> > >
> > > Thank you for your answers.

---

### Official Review · Reviewer_QPpV · 2021-11-01

**Correctness:** 4
**Technical Novelty And Significance:** 3
**Empirical Novelty And Significance:** 3
**Recommendation:** 8
**Confidence:** 4

**Main Review:**

Strengths:
1. The authors provide both intuition and theoretical explanations of the proposed method supported by experimental results.
2. Although all components are widely used techniques in the field, the application of RL to tackle pseudo-labeled sample selection for keypoint localization is novel.
3. The proposed method achieves the improvement upon the state-of-the-art on several benchmarks for human and animal body landmark localization and specifically in a low labeled data regime (5% of data is labeled).
4. Informative ablation studies and evaluation of the generalization ability on domain transfer.

Weaknesses:
1. The RL part of the approach has many moving parts and it would be beneficial to justify the choices of hyperparameters in the proximal policy optimization algorithm.
2. Inner-loop network training is executed multiple times during policy learning. How has the training complexity (time to convergence) increased compared to previous works?

Questions:
1. Section 3.2: Is Eq(1) computed per image or per keypoint given that each image has K keypoints? If an image has some keypoints with the confidence above a threshold and some are below, does the image get selected for the training round?
2. How does the proposed pseudo-labeling strategy deal with not-visible keypoints in the image (e.g., a left eye and a left-wing of a bird are not visible if the bird is depicted from the right-side viewpoint)?


**Summary Of The Paper:**

The paper introduces a method for semi-supervised keypoint localization based on pseudo-labeling with auto-curriculum learning. The Auto-curriculum learning approach learns a series of dynamic thresholds for automatic selection of high-quality pseudo-labeled examples for model retraining. The reinforcement learning (RL) framework, more specifically, the proximal policy optimization algorithm, is used to search for the optimal curriculum. The method is evaluated on four benchmarks in keypoint localization.

**Summary Of The Review:**

The paper is well-written and easy to follow. The proposed method is described in detail. The method is evaluated on four popular datasets for the keypoint localization task. The proposed method demonstrates superior performance especially in cases with only 5% of labeled data (out of no more than 10,000 examples). Ablation studies justify the design choices. The method, while combining existing techniques, is proven experimentally to be superior to the previous works and will add to the body of knowledge on keypoint localization.

---

> ### Author Response · Authors · 2021-11-20
> **To Reviewer QPpV**
>
> We thank the reviewer for the insightful feedback and are excited about the positive comments on our paper.
>
>
> **Q1: The RL part of the approach has many moving parts and it would be beneficial to justify the choices of hyperparameters in the proximal policy optimization algorithm.**
>
> A1: Most hyperparameters are default settings of PPO2 (Schulmanet al., 2017). The only parameters we changed are $T$ (sampling steps) and $M$ (number of sampled curricula in each step).
> In experiments, we use $T=16$ and $M=8$ for RL curriculum search. Empirically, we find that our PLACL is not sensitive to these hyperparameters. For example, in the LSPET dataset with 5% labeled data, we have tried using larger $T$ and $M$ ($T=32$ and $M=16$). The results are $70.76$ PCK for $T=16$ and $M=8$, versus $70.84$ PCK for $T=32$ and $M=16$.
>
>
> **Q2: Inner-loop network training is executed multiple times during policy learning. How has the training complexity (time to convergence) increased compared to previous works?**
>
> A2: Although the RL search process indeed increases the training complexity, the total training cost is not too high (only 1.5 days with 32 NVIDIA Tesla V100 GPUs).
>
> There are a lot of methods that target at improving the searching efficiency in literature, *e.g.* sample-efficient reinforcement learning [1], and reduced-training proxy tasks (input size, model size, training samples, and training epochs can be reduced) [2].
> With these techniques, we are able to get orders of magnitude less computation cost, but can still match the performance. Therefore, we believe that the scalability is NOT a problem.
> For example, in order to further reduce the complexity, we use a light proxy task (with reduced training samples) for the RL search process. Specifically, we randomly select a small proportion of the training data (*e.g.* 5k images) for efficient curriculum search. After the search procedure, we re-train the keypoint networks with the searched curriculum on the full training set and evaluate them on the test set.
>
> As shown in Table R1 , we randomly select different number of training images (5k and 10k) for RL curriculum search. We find that reducing the number of training images by half (from 10k to 5k) does not decrease the final performance much, which validates the effect of using proxy tasks.
> Such a strategy enables us to apply the proposed PLACL approach to large-scale datasets, such as MS-COCO'2017 and full MPII dataset.
>
> **Table R1: We perform RL curriculum search on different numbers of training images (\#Images), and re-train the keypoint networks with the searched curriculum on the full training set.**
>
> | Dataset    | #Images | 5% | 10% | 20% | 50% |
> | :--------  |  :--:   | :--: |  :--: | :--: | :--: |
> | LSPET      | 5k      | 70.72$\pm$1.49 | **71.93$\pm$1.17** | 72.24$\pm$0.82 | 72.71$\pm$1.19 |
> | LSPET      | 10k     | **70.76$\pm$1.47**  | 71.91$\pm$1.15  | **72.30$\pm$0.88**  | **72.73$\pm$1.23**  |
> |MS-COCO'2017| 5k      | **69.39$\pm$1.03**  | 70.11$\pm$0.89  | **71.84$\pm$0.66**  | 73.42$\pm$0.57  |
> |MS-COCO'2017| 10k     | 69.24$\pm$1.02 | **70.12$\pm$0.87**  | 71.61$\pm$0.63 | **73.43$\pm$0.61** |
> | | |
>
> [1] Buckman, Jacob, et al. "Sample-efficient reinforcement learning with stochastic ensemble value expansion." arXiv preprint arXiv:1807.01675 (2018).
>
> [2] Zhou, Dongzhan, et al. "Econas: Finding proxies for economical neural architecture search." Proceedings of the IEEE/CVF Conference on Computer Vision and Pattern Recognition. 2020.
>
>
>
> **Q3: Section 3.2: Is Eq(1) computed per image or per keypoint given that each image has K keypoints? If an image has some keypoints with the confidence above a threshold and some are below, does the image get selected for the training round?**
>
> A3:  Eq(1) is computed on the keypoint level. If an image has some keypoints with the confidence above a threshold and some are below, those confident keypoints will be selected for training.
>
>
>
> **Q4: How does the proposed pseudo-labeling strategy deal with not-visible keypoints in the image (e.g., a left eye and a left-wing of a bird are not visible if the bird is depicted from the right-side viewpoint)?**
>
> A4:  We deal with visible and not-visible keypoints in a unified manner without special treatment. However, for these not-visible keypoints, the confidence scores of the pseudo-labels are usually lower. Low-confidence pseudo-labels will not be selected for model training.

---

> > ### Comment · Reviewer_QPpV · 2021-11-30
> > **Responce**
> >
> > Thank you for responding to my concerns and questions.

---

### Official Review · Reviewer_6dEq · 2021-11-04

**Correctness:** 3
**Technical Novelty And Significance:** 3
**Empirical Novelty And Significance:** 3
**Recommendation:** 6
**Confidence:** 3

**Main Review:**

##########################################################################

Strengths:
1. The task is practical and motivation is well. How to select the threshold of pseudo label is an important and complicated task.

2. Overall, the method of this paper is technically reasonable and novel. The paper first applies curriculum learning to semi-supervised keypoint localization and proposes use RL to search the best curriculum.

3. The experiments show that the proposed method is effective compared to other SSL methods. The ablation study validates the three parts of proposed method are all important and can improve performance (about 2%-5%).



##########################################################################

Weaknesses:
1. In my opinion, the time complexity of proposed algorithm is too high. The RL search process increases the training time by T*M times (=128 in this paper). The high complexity will make the model less scalable, such as in dataset size. The author maybe can explain about the current training cost and whether the scalability is indeed a problem.

2. The dataset in experiments in the article is somewhat simple and small in scale. If the mainstream datasets in human pose estimation such as COCO keypoint, or full MPII, can be used in experiments, the contributions would be more convincing.

3. The experimental comparison in Effect of parameter search is not sufficient, and the selected baseline method is weak.
The paper only compares Random Search and does not fully explain the details of this baseline. Does this baseline method also select the optimal strategy from candidates of the same size (T*M)?
In addition, considering that using RL to search is one of the main contributions of this paper, the author should consider comparing other stronger and more comprehensive baselines. For example, manually design many curriculums whose thresholds are gradually decreasing (on the epoch level) and select the curriculum with the best result.




##########################################################################

Minor comments:
1. The division and use of data sets are not particularly clear. Are the reported numbers all evaluated in testing set, and D_val in Eq.3 uses validation set? But the paper says that the MPII validation set is for evaluation. So what is the D_val in MPII.

2. In the paper, the symbol N represents both the keypoint network and the number of epochs, which is somewhat confusing.

3. In my opinion, the proposed method is not particularly related to the keypoint localization task. It will be better if this method can be applied to other tasks.

4. In addition, the author maybe can add discussion on the following papers about semi-supervised human pose estimation.
Rongchang Xie, Chunyu Wang, Wenjun Zeng, Yizhou Wang. An Empirical Study of the Collapsing Problem in Semi-Supervised 2D Human Pose Estimation. ICCV 2021.

##########################################################################

Updates:

Thanks for the authors' response. The response and new resutls address my main concern. I tend to accept this paper.



**Summary Of The Paper:**

1. This paper introduces Curriculum Learning to semi-supervised keypoint localization, which is an automatic pseudo-labeled data selection method. The method uses reinforcement learning to learns a series of dynamic thresholds.

2. Besides, this paper proposes the cross-training strategy for pseudo-labeling to alleviate confirmation bias.

3. The experiments shows that the proposed method can effectively improve the performance in different dataset and surpass other semi-supervised methods.


**Summary Of The Review:**

This paper proposes a novel and effective threshold selection method for semi-supervised keypoint localization. Meanwhile, I think there are some weaknesses in practicality. I currently choose borderline accept. The author may can explain about the weaknesses.

---

> ### Author Response · Authors · 2021-11-20
> **To Reviewer 6dEq (Part1)**
>
> Thank you for the insightful feedback and appreciating our technical contributions and empirical evaluation. We have addressed your concerns and conducted additional experiments. The paper is revised accordingly with all the requested results added.
>
> **Q1: Concerns about time complexity**
>
> Although the RL search process indeed increases the training complexity, the total training cost is not too high (only 1.5 days with 32 NVIDIA Tesla V100 GPUs).
>
> There are a lot of methods that target at improving the searching efficiency in literature, *e.g.* sample-efficient reinforcement learning [1], and reduced-training proxy tasks (input size, model size, training samples, and training epochs can be reduced) [2].
> With these techniques, we are able to get orders of magnitude less computation cost, but can still match the performance. Therefore, we believe that the scalability is NOT a problem.
> For example, in order to further reduce the complexity, we use a light proxy task (with reduced training samples) for the RL search process. Specifically, we randomly select a small proportion of the training data (*e.g.* 5k images) for efficient curriculum search. After the search procedure, we re-train the keypoint networks with the searched curriculum on the full training set and evaluate them on the test set.
>
> As shown in Table R1 , we randomly select different number of training images (5k and 10k) for RL curriculum search. We find that reducing the number of training images by half (from 10k to 5k) does not decrease the final performance much, which validates the effect of using proxy tasks.
> Such a strategy enables us to apply the proposed PLACL approach to large-scale datasets, such as MS-COCO'2017 and full MPII dataset.
>
> **Table R1: We perform RL curriculum search on different numbers of training images (\#Images), and re-train the keypoint networks with the searched curriculum on the full training set.**
>
> | Dataset    | #Images | 5% | 10% | 20% | 50% |
> | :--------  |  :--:   | :--: |  :--: | :--: | :--: |
> | LSPET      | 5k      | 70.72$\pm$1.49 | **71.93$\pm$1.17** | 72.24$\pm$0.82 | 72.71$\pm$1.19 |
> | LSPET      | 10k     | **70.76$\pm$1.47**  | 71.91$\pm$1.15  | **72.30$\pm$0.88**  | **72.73$\pm$1.23**  |
> |MS-COCO'2017| 5k      | **69.39$\pm$1.03**  | 70.11$\pm$0.89  | **71.84$\pm$0.66**  | 73.42$\pm$0.57  |
> |MS-COCO'2017| 10k     | 69.24$\pm$1.02 | **70.12$\pm$0.87**  | 71.61$\pm$0.63 | **73.43$\pm$0.61** |
> | | |
>
> [1] Buckman, Jacob, et al. "Sample-efficient reinforcement learning with stochastic ensemble value expansion." arXiv preprint arXiv:1807.01675 (2018).
>
> [2] Zhou, Dongzhan, et al. "Econas: Finding proxies for economical neural architecture search." Proceedings of the IEEE/CVF Conference on Computer Vision and Pattern Recognition. 2020.
>
>
> **Q2: If the mainstream datasets in human pose estimation such as COCO keypoint, or full MPII, can be used in experiments, the contributions would be more convincing.**
>
> A2: Thanks for the suggestions! We have added the experiments on COCO and full MPII as suggested. Since the codes of SSKL are not publicly available, we only compare with CL in the experiments. The results in Table R2 below show that our proposed PLACL consistently outperforms CL on these mainstream datasets. Note that our experimental results on COCO and full MPII are obtained using reduced-training proxy tasks, *i.e.* we use 5K images for RL curriculum search, and re-train the model with the obtained curriculum on the full training set.
>
> **Table R2: Comparisons with CL on the MS-COCO'2017 and full MPII datasets.**
>
> | Dataset | Method   | 5% | 10% | 20% | 50% | 100% |
> | :--------  |  :--:   | :--: |  :--: | :--: | :--: |  :--:  |
> | MS-COCO'2017 | HRNet  | 62.44$\pm$1.26 | 66.02$\pm$1.07 | 69.62$\pm$0.84 | 72.81$\pm$0.73 | 74.61$\pm$0.58 |
> | MS-COCO'2017 | CL  | 64.47$\pm$1.18  | 67.82$\pm$0.95 | 70.36$\pm$0.89 | 72.92$\pm$0.84 |
> | MS-COCO'2017 | PLACL (Ours) | **69.39$\pm$1.03**  | **70.11$\pm$0.89**  | **71.84$\pm$0.66**  | **73.42$\pm$0.57**  |
> | FULL MPII | HRNet| 78.00$\pm$1.35  | 81.89$\pm$0.94 | 82.94$\pm$0.67 | 88.34$\pm$0.45 | 89.76$\pm$0.17  |
> | FULL MPII | CL  | 80.38$\pm$1.31 | 83.06$\pm$0.89 | 84.57$\pm$0.68 | 88.72$\pm$0.34 |
> | FULL MPII | PLACL (Ours) | **82.21$\pm$1.22**  | **85.42$\pm$0.85**  | **86.24$\pm$0.56**  | **89.16$\pm$0.21**  |
> | | |
>
> **Q3.1: The paper only compares Random Search and does not fully explain the details of this baseline. Does this baseline method also select the optimal strategy from candidates of the same size ($T \times M$)?**
>
> A3.1: Yes. We randomly sampled $T \times M$ curricula. Each curriculum consists of a series of random thresholds for pseudo-labeled data selection. We pick out the best one as the ''PLACL, w/o PPO2 search'' baseline. Our proposed PLACL achieves much better performance than the Random Search baseline. This suggests that searching the optimal thresholds is non-trivial and PPO2 is effective in improving the search efficiency.

---

> > ### Author Response · Authors · 2021-11-20
> > **To Reviewer 6dEq (Part2)**
> >
> > **Q3.2: The author should consider comparing other stronger and more comprehensive baselines. For example, manually design many curriculums whose thresholds are gradually decreasing (on the epoch level) and select the curriculum with the best result.**
> >
> >
> > A3.2: Thanks for the valuable suggestion! As suggested, we have added the baselines whose thresholds are gradually decreasing on the epoch level. We tried five curricula with different decrease slopes and reported the best one in the Table R3 below. With the thresholds decreasing, the number of selected samples gradually increases. Kindly note that this baseline is also similar to the CL (Cascante-Bonilla et al., 2021) method, where the curriculum is also manually designed, but the thresholds are gradually decreasing on the *round* level. We observe that PLACL significantly outperforms the manually-designed curriculum baseline which validates the effectiveness of automatic curriculum search.
> >
> > **Table R3: Comparison with manually designed curriculum on LSPET dataset with 5% labeled data.**
> >
> > | Method   | PCK\@0.1|
> > | :--------  |  :--:   |
> > | Decreasing Thresholds |  65.71 |
> > | PLACL |  70.76 |
> > | | |
> >
> > **Q4: The division and use of data sets are not particularly clear. Are the reported numbers all evaluated in testing set, and $\mathbb{D}_\text{val}$ in Eq.3 uses validation set? What is the $\mathbb{D}_\text{val}$ in MPII?**
> >
> > A4: (1) For fair comparisons, we use the same settings as SSKL (Moskvyak et al., 2020). For LSPET dataset, We use 10,000 images from the extended version (Johnson \& Everingham, 2011) for training and 2,000 images from the original version (Johnson \& Everingham, 2010) for validation and testing. For MPII dataset, we use 10,000 random images from MPII `train` for training, and another 3,311 images from MPII `train` for validation and MPII `val` for testing. For CUB-200-2011 dataset, we split the dataset into training (100 categories with 5,864 images), validation (50 categories with 2,958 images) and testing (50 categories with 2,966 images). For ATRW dataset, we use the official dataset splits with 3,610 images for training, 516 images for validation, and 1,033 images for testing. For AnimalPose dataset, we use the official dataset splits with 2,798 images for training, 810 images for validation, and 1,000 images for testing.
> >
> > (2) Yes. All the reported numbers are evaluated in our testing set, and $\mathbb{D}_\text{val}$ in Eq.3 uses the validation set.
> >
> > (3) The $\mathbb{D}_\text{val}$ in MPII is the randomly selected 3,311 images from the MPII `train`, which does not intersect with the 10,000 training images.
> >
> >
> >
> >
> > **Q5: The symbol N represents both the keypoint network and the number of epochs, which is somewhat confusing.**
> >
> > A5: Thanks! The keypoint network is now represented by $\Theta$ in the revised version.
> >
> > **Q6: In my opinion, the proposed method is not particularly related to the keypoint localization task. It will be better if this method can be applied to other tasks.**
> >
> > A6: Great suggestion! We plan to investigate the applicability of PLACL to other tasks, such as object detection and semantic segmentation in the future.

---

### Decision · Program_Chairs · 2022-01-20

**Decision:**

Accept (Poster)

**Comment:**

This paper proposes a pseudo-labeled data selection method for semi-supervised pose estimation. The investigated task in this paper is practical and useful. The framework is well designed and reasonable, and extensive ablation studies are conducted to test the efficacy of the method. After discussion, all the reviewers recommend accept of this paper.